# Characterizing changes in behaviors associated with chemical exposures during the COVID-19 pandemic

**Julie B. Herbstman**[1◉]*, **Megan E. Romano**[2◉], **Xiuhong Li**[3], **Lisa P. Jacobson**[3], **Amy E. Margolis**[4], **Ghassan B. Hamra**[3], **Deborah H. Bennett**[5], **Joseph M. Braun**[6], **Jessie P. Buckley**[7], **Trina Colburn**[8], **Sean Deoni**[9], **Lori A. Hoepner**[10], **Rachel Morello-Frosch**[11], **Kylie Wheelock Riley**[12], **Sheela Sathyanarayana**[13], **Susan L. Schantz**[14], **Leonardo Trasande**[15], **Tracey J. Woodruff**[16], **Frederica P. Perera**[1◉], **Margaret R. Karagas**[2◉], on behalf of program collaborators for Environmental influences on Child Health Outcomes[¶]

1 Department of Environmental Health Sciences, Columbia University Mailman School of Public Health, New York, NY, United States of America, 2 Department of Epidemiology, Dartmouth College Geisel School of Medicine, Hanover, NH, United States of America, 3 Department of Epidemiology, Johns Hopkins University Bloomberg School of Public Health, Baltimore, MD, United States of America, 4 Department of Psychiatry, Columbia University Irving Medical Center, New York, NY, United States of America, 5 Department of Public Health Sciences, University of California—Davis, Davis, CA, United States of America, 6 Department of Epidemiology, Brown University School of Public Health, Providence, RI, United States of America, 7 Department of Environmental Health Sciences, Johns Hopkins University Bloomberg School of Public Health, Baltimore, MD, United States of America, 8 Department of Child Health, Behavior, and Development, Seattle Children's Hospital, Seattle, WA, United States of America, 9 Department of Pediatrics, Rhode Island Hospital, Providence, RI, United States of America, 10 Department of Environmental Health Sciences, Columbia University Mailman School of Public Health and Department of Environmental and Occupational Health Sciences, SUNY Downstate Health Sciences University School of Public Health, New York, NY, United States of America, 11 Department of Environmental Health Sciences, University of California—Berkeley, Berkeley, CA, United States of America, 12 Department of Environmental Health Sciences, Columbia University Mailman School of Public Health, Health, New York, NY, United States of America, 13 Department of Pediatrics, University of Washington and Seattle Children's Research Institute, Seattle, WA, United States of America, 14 Department of Comparative Biosciences, University of Illinois—Urbana-Champaign, Champaign, IL, United States of America, 15 Departments of Pediatrics and Population Health, New York University Grossman School of Medicine, New York, NY, United States of America, 16 Department of Obstetrics and Gynecology, University of California San Francisco School of Medicine, San Francisco, CA, United States of America

◉ These authors contributed equally to this work.
¶ Membership of the Environmental influences on Child Health Outcomes is listed in the Acknowledgments.
* jh2678@cumc.columbia.edu

**Data Availability Statement:** The data used in this analysis are maintained by the ECHO Data Analysis Center, from which access may be requested. Per

## Abstract

The COVID-19 pandemic—and its associated restrictions—have changed many behaviors that can influence environmental exposures including chemicals found in commercial products, packaging and those resulting from pollution. The pandemic also constitutes a stressful life event, leading to symptoms of acute traumatic stress. Data indicate that the combination of environmental exposure and psychological stress jointly contribute to adverse child health outcomes. Within the Environmental influences on Child Health Outcomes (ECHO)-wide Cohort, a national consortium initiated to understand the effects of environmental exposures on child health and development, our objective was to assess whether there were pandemic-related changes in behavior that may be associated with

ECHO policies, individuals requesting to use the data should contact ECHO-DAC@rti.org.

**Funding:** Research reported in this publication was supported by the Environmental influences on Child Health Outcomes (ECHO) program, Office of The Director, National Institutes of Health, under Award Numbers U2COD023375 (Coordinating Center), U24OD023382 (Data Analysis Center), U24OD023319 (PRO Core), and UH3OD023290 (Columbia University, New York, New York: Perera FP, Herbstman JB); UH3OD023275 (Dartmouth College, Hanover, New Hampshire: Karagas MR); UH3OD023272 (University of Illinois, Urbana: Schantz SL, University of California, San Francisco: Woodruff T, University of California, Berkeley, Morello-Frosch, R); UH3OD023271 (University of Washington, Seattle: Karr C, Sathyanarayana S); and UH3OD023313 (Memorial Hospital of Rhode Island, Pawtucket: Deoni S, D'Sa VA; Brown University, Providence,RI: Braun J). The funders had no role in study design, data collection and analysis, decision to publish, or preparation of the manuscript.

**Competing interests:** The authors have declared that no competing interests exist.

environmental exposures. A total of 1535 participants from nine cohorts completed a survey via RedCap from December 2020 through May 2021. The questionnaire identified behavioral changes associated with the COVID-19 pandemic in expected directions, providing evidence of construct validity. Behavior changes reported by at least a quarter of the respondents include eating less fast food and using fewer ultra-processed foods, hair products, and cosmetics. At least a quarter of respondents reported eating more home cooked meals and using more antibacterial soaps, liquid soaps, hand sanitizers, antibacterial and bleach cleaners. Most frequent predictors of behavior change included Hispanic ethnicity and older age (35 years and older). Respondents experiencing greater COVID-related stress altered their behaviors more than those not reporting stress. These findings highlight that behavior change associated with the pandemic, and pandemic-related psychological stress often co-occur. Thus, prevention strategies and campaigns that limit environmental exposures, support stress reduction, and facilitate behavioral change may lead to the largest health benefits in the context of a pandemic. Analyzing biomarker data in these participants will be helpful to determine if behavior changes reported associate with measured changes in exposure.

## Introduction

As a result of COVID-19 lockdowns and restrictions, there have been many changes in behaviors and possibly in many environmental exposures associated with those behaviors. For example, air pollution levels were initially lower in many places as a result of work-from-home policies and school closures [1–3]. Conversely, exposure to cleaning and disinfection products may have increased due to enhanced efforts to clean hands and surfaces to reduce viral spread [4–6], particularly towards the beginning of the pandemic due to concern about fomites, and before person-to-person airborne exposure was widely recognized as the primary source of COVID-19 transmission. These changes are dynamic and result in 'natural experiments' to understand how changing exposures influence health outcomes that are sensitive to environmental influence. Prenatal and early life exposure to air pollutants can increase the risk of adverse health effects including preterm birth, lower birth weight, shortened gestational age, asthma incidence and exacerbation, and neurodevelopmental outcomes [7, 8]. Likewise exposure to endocrine-disrupting chemicals (e.g., phenols, phthalates, and parabens) found in personal care and cleaning products and processed foods/food packaging have been found to adversely influence birth outcomes and physical growth [9] and may further influence child respiratory health[10–14] and neurodevelopmental outcomes [15, 16]. Thus, an unintended consequence of pandemic-related changes in behavior may influence a wide range of environmental exposures that have implications for aspects of health aside from COVID-19.

The Centers for Disease Control and Prevention (CDC) has recommended cleaning and disinfecting as best practices to reduce transmission both in households and public places [17]. Cleaning and disinfection products encompass reactive compounds such as phenols, chloride, ammonia, hydrogen peroxide and other acids and compounds, though these compounds may not be necessary for effective disinfection [18]. Additionally, hand hygiene, such as washing hands or use of hand sanitizers has been encouraged to prevent SARS-Cov-2 infection. In addition to active compounds such as alcohol in hand sanitizers, soaps and sanitizers often include fragrance which incorporates phthalates as scent retainers.

Measures that include limiting physical interactions, restaurant closures, and reduced hours for grocery stores or food markets during the COVID-19 pandemic in concert with impacts on the food supply chain have likely led to changes in dietary habits during the pandemic [19, 20]. One consumer survey suggests that 72% of individuals are shopping less frequently for food. Among families, 70% indicate that they are snacking more frequently, and 88% reported an increased number of meals prepared at home [21]. Increased consumption of canned goods or processed food may lead to greater exposure to a variety of endocrine-disrupting chemicals including bisphenol A (BPA, and its substitutes), phthalates, and perfluoroalkyl substances [22–26]. Conversely, eating less fast food meals may reduce exposure to some of these same chemicals found in food and food packaging [27, 28]. Therefore, in some cases, pandemic-related behavior changes may increase chemical exposures and in other cases, may reduce exposure.

Women of reproductive age are among the most common consumers of personal care products and cosmetics [29]. Stark differences exist in the types and quantities of products used across racial/ethnic groups; African Americans purchase nine times more beauty products than Non-Hispanic Whites, and Latinas are the fastest growing demographic in ethnic beauty product market [29]. Chemicals found in these products include phthalates (e.g., diethyl phthalate (DEP), di-*n*-butyl phthalate (DnBP) and di-isobutyl phthalate (DiBP)), parabens and other environmental phenols that have endocrine disrupting properties [10, 30].

Importantly, public health agencies, including the CDC, acknowledge that the COVID-19 pandemic can increase stress. A recent review of the literature documented a 16–28% increase in anxiety and depression and an 8% increase in self-reported stress associated with the COVID-19 pandemic, and that individual and structural factors may moderate risk of these symptoms [31]. In addition to depression and anxiety, school closures have led to mental distress among children, with indication that people from marginalized communities may have a worse experience [32]. Finally, prior studies document higher rates of psychological distress and posttraumatic stress symptoms after a quarantine due to exposure to an infectious disease [33].

The Environmental influences on Child Health Outcomes (ECHO)-wide Cohort is well-positioned to address the interaction between environmental and psychosocial stressors [34]. This consortium-based cohort has representation from 69 U.S. based cohort studies who began using a common data collection protocol in 2019. As of March 4, 2022, the consortium includes 29,622 children who have been consented to new data collection under the common protocol. These children were born in 672 counties from 49 states, the District of Columbia and Puerto Rico; however, it is not a probability-based sample that can be generalized to the U.S. population.

We hypothesized that COVID-19 would result in changes in environmental chemical exposures and pandemic-related traumatic stress. Here, our objective was to understand how behaviors related to environmental exposures have changed in conjunction with the COVID-19 pandemic and whether pandemic-related traumatic stress is associated with pandemic-related behavior change. Future biomarker analyses can determine whether behavior changes observed here associated with chemical exposure levels.

## Methods

The ECHO-wide Cohort Study is an NIH-funded research collaborative [35], in which 69 ongoing cohorts contributed extant data and enrolled participants for continued data collection according to an IRB-approved ECHO-wide Cohort Protocol (EWCP). Participants were initially enrolled at various life stages from pre-conception through early childhood, with the

majority recruited during pregnancy or at birth. Cohorts were asked to administer a new survey to elicit information on environmental exposures and subsequent behaviors during the COVID-19 pandemic from pregnant women or caregivers of children enrolled in the study. The data were managed, harmonized and analyzed by the ECHO Data Analysis Center in a Federal Information Security Modernization Act (FISMA) moderate platform [35].

## Changes in Environmental Exposures (CEE) component of covid-19 survey

A brief series of questions capturing data on COVID-19 Changes in the Environmental Exposures (CEE) (see *S1 File*) was developed to focus on behaviors that have been previously associated with exposure to environmental chemicals, including diet, food preparation, personal care product and consumer product use. The survey was programmed in REDCAP Central, a centralized web-based data entry system, to be administered along with the follow-up COVID-19 questionnaire, which elicited information about COVID-related acute stress (see below). Both the CEE survey and COVID-19 questionnaires were available in both English and Spanish, and could be either interviewer-or self-administered. Administration began after the ECHO-wide Cohort Central IRB approval in December 2020 and local IRB approval (as necessary) was obtained (December 2020 to April 2021). Nine ECHO cohorts located in New Hampshire, Rhode Island, New York, Illinois, California, and Washington elected to administer this survey. A total of 1535 participants completed the survey from December 2020 through May 2021. Although data collection is still ongoing, more than 67% of the data reported here were contributed by participants in the rural state of New Hampshire by the New Hampshire Birth Cohort Study. This differential response was due to variation in the underlying cohort sample sizes and protocol procedures, including IRB approval, that enabled faster/slower roll-out of survey administration by cohort.

## Pandemic-related Traumatic Stress (PTS) symptoms measured in the COVID-19 survey

Ten items were administered to measure PTS symptoms [36] based on acute stress disorder symptoms measured in the DSM-5 [37], which evaluate symptomology and severity within the month following a traumatic event. Although living during a pandemic may not be traditionally viewed as an inciting event in the DSM-5 (e.g., post-traumatic stress disorder (PTSD)) [37], the COVID-19 pandemic did present as a life-threatening experience–both real and perceived–to many individuals. COVID-19 has been described as eliciting traumatic stress reactions above and beyond distress related to disruptions to daily life [38, 39]. It has infected nearly 80 million people in the US, taken the lives of around 1 million Americans and has been responsible for an estimated 900,000 hospitalizations, of which 20% required ICU intervention, and causing long-term physical, mental, and cognitive consequences in one in three survivors [40, 41]. The fear and experiences of contracting the virus, witnessing or putting a loved one at risk, and suffering severe illness or death all reflect the life-threatening nature of the pandemic that can result in traumatic stress reactions [41, 42]. The survey was administered concurrently with the CEE survey, well after the onset of the COVID-19 pandemic (provided in S1 File). We thus further conjecture that given the protracted nature of the COVID-19 pandemic, this scale captures a relevant measure of pandemic-related traumatic stress rather than acute stress per se. Regardless of timing, the scale captures an individual's experience of acute stress symptoms in the context of the pandemic.

Given time limitations and concerns about participant burden, 10 items were developed to query these 5 areas: *intrusion* (e.g., distressing dreams, been distressed when seeing something that is reminiscent of COVID-19); *negative mood* (e.g., anhedonia or the inability to feel

pleasure, anger disproportionate to the situation); *disassociation* (e.g., feelings of time slowing); *avoidance* (e.g., purposeful efforts to avoid thinking about the event or actions that are not congruent with required realities of persisting threats); and difficulty regulating *arousal* (e.g., sleep disturbance, irritability, poor concentration). Questions began "Since becoming aware of the COVID-19 outbreak, how often have you [experienced the following]". Each item was assessed using a Likert scale: not at all (1), rarely (2), sometimes (3), often (4), very often (5). An endorsement of 3 (sometimes), 4 (often), or 5 (very often) is considered significant. The Total number of symptoms was calculated as a sum of those symptoms (items) endorsed at a significant level (endorse at level 3, 4, or 5) [36].

## Statistical approach

We report descriptive statistics of cross-sectionally obtained survey data on the reported changes in environmental exposures and behaviors from pre-COVID to the COVID era. Data from the CEE form were merged with other ECHO-Wide Cohort Protocol (EWCP) demographic and geographic data to examine whether these factors along with COVID-related stressors are associated with pandemic-related behavior change. Because the pandemic created dynamic circumstances that may influence behaviors under investigation, we compared those who responded early in the survey administration period (e.g., December 2020 through February 2021) and those who responded later in the survey administration period (e.g., March 2021 through May 2021); this cut-point distinguishes those who responded during the first year of the pandemic vs. afterwards.

This survey was formatted so that for each question, respondents had the option to indicate whether their behaviors increased, decreased or did not change. Because we focused our analysis on outcomes that varied (either increased or decreased), behaviors were coded as binary, based on how the data were collected (e.g., more or less than before the pandemic). These outcomes include: eating more home cooked meals, eating less fast food, drinking more alcoholic beverages, drinking less alcoholic beverages, using more antimicrobial soaps, using more bleach products, and using more hand sanitizer. Anyone who reported using more items of interest was defined as yes (= 1) for the 'more' outcome variables, the remaining participants (those who reported the same or less) were defined as no (= 0) for the same outcome behavior. Similarly, the less outcome was defined as doing the behavior less than before the pandemic vs. all others (i.e., same and more).

Composites were created for outcome variables where the outcome was characterized by multiple individual components. We calculated whether participants use fewer hair products if reported less use of at least 2 products (perms, hair dye, hair spray, hair gel), and use less make-up/body products if reported less use of at least 2 products (nail polish, make-up, perfume). The ultra-processed foods include 10 food categories based on the NOVA classification system, a validated method by which all foods are grouped based on the industrial processes they undergo [43]: sweetened milk substitute, sweetened yogurt or ice cream, sweetened beverages, commercially made desserts, frozen meats, meatless patties, commercially-made breads, dry cake mixes, packaged snacks, and sweetened breakfast cereals. For each participant, the total number of more food items that were eaten (sum score ranged 0–10) and less food items eaten (sum score ranged 0–10) were created first, and then the top 10% of the sum scores of the ultra-processed food, (e.g., > = 3 of eating more ultra-processed food, and > = 3 of eating less ultra-processed food), were used as cutoffs to define these binary outcomes.

Chi-square tests were used to examine whether a specific outcome varied by geographic location, demographic characteristics, or stress from the COVID-19 pandemic. Separate logistic regression models were run for each outcome to examine the association between possible

factors and behavioral change outcomes due to the COVID-19 pandemic. We used two multi-variable logistic models, where model 1 included age, race/ethnicity (Hispanic, non-Hispanic White, non-Hispanic Black, non-Hispanic Asian, and non-Hispanic Other), education, employment, marital status, number of people in the household, whether any household member tested COVID-19 positive, and whether the cohort was New Hampshire (NH) vs. non-NH. In addition to these characteristics, model 2 included the number of stress symptoms (none (0), minimal (1–3), moderate (4–6), and severe (7–9) stress symptoms) as a predictor of the behaviors. We also examined whether there was a monotonic trend of the number of pandemic-related traumatic stress symptoms associated with each outcome after adjusting for all variables in model 1.

Missing data were addressed in the analyses with multiple imputation (MI) by fully conditional specification with a discriminant function [44]. Imputation was performed for: respondents' age (percentage missing was 6.3%), race/ethnicity (4.6%), maternal education (16.0%), employment (4.4%), marital status (27.4%), number of people living in the household (0.8%), whether there was a COVID-19 positive member of the household (3.5%), total number of stress symptoms (7.6%), and all outcome variables, i.e., eat more home cooked meals (0.8%), less fast food (1.0%), less alcoholic beverages (2.8%), more for $\geq 3$ processed foods (6.7%), less for $\geq 5$ processed food (6.7%), less hair products (5.0%), less make-up/body products (2.9%), more antimicrobial soaps (0.5%), more hand sanitizer (0.5%), more antibacterial household cleansers (1.3%), and more bleach products (1.4%). Imputation models included all variables listed above and cohort group membership (NH vs non-NH).

# Results

## Study participants and pandemic context

A total of 1535 individuals responded to the survey; 11 participants responded more than once however only their first response was included here. They had children participating in the ECHO-wide cohort between the ages of 0–17.9 years (median 5.9, IQR 2.26–8.49). The median age of survey respondents was 37 years. In addition, 117 (8%) of respondents were pregnant when they completed the survey. Among respondents, 77% completed the survey from December 2020 through February 2021, and 23% responded from March 2021 through May 2021 (**Table 1**). Generally, respondents were similar across these two periods, except with respect to ethnicity. Those who participated in the earlier window were more likely (87%) to self-identify as Non-Hispanic White (vs. 45% in the later window) while 32% of respondents who reported in the later window self-identified as Hispanic (vs. 6% in the earlier window). This is mainly due to differential cohort protocols with respect to survey administration, such that approximately two-thirds of respondents included in this analysis were from the New Hampshire area. Subsequent analyses did not distinguish between the response periods; however, we considered these ethnic differences within the response periods and covaried for cohort group membership in the interpretation of results.

Because the survey administration spanned December 2020 through May 2021, we inquired about local governmental advice, personal and household COVID-19 infection positivity, and stress symptoms associated with the pandemic at the time of survey completion. At the time of survey completion, 43% of all respondents indicated that their local government was encouraging them to stay at home and 41% recommended social distancing when going out; 12% were ordered by their local government to stay at home. Of all respondents, 10% indicated that they or a member of their household had tested positive for COVID-19 prior to survey completion. Among survey respondents, 13%, 38%, 36%, and 14% reported experiencing no,

**Table 1. Characteristics of respondents to the CEE survey December 2020-May 2021.**

| Characteristics, N(%) with data | December 2020—February 2021 | March—May 2021 | Overall |
|---|---|---|---|
| **# of participants** | **1177** | **358** | **1535** |
| **Age of respondent (years)** | **1145(97%)** | **293(82%)** | **1438 (94%)** |
| 19–24 years | 13(1%) | 26(9%) | 39(3%) |
| 25–34 years | 377(33%) | 107(37%) | 484(34%) |
| 35–44 years | 659(58%) | 134(46%) | 793(55%) |
| 45+ years | 96(8%) | 26(9%) | 122(8%) |
| **Race/Ethnicity,** | **1127(96%)** | **337(94%)** | **1464 (95%)** |
| Hispanic | 71(6%) | 108(32%) | 179(12%) |
| non-Hispanic-White | 983(87%) | 150(45%) | 1133 (77%) |
| non-Hispanic-Black | 12(1%) | 23(7%) | 35(2%) |
| non-Hispanic-Asian | 5(<1%) | 20(6%) | 25(2%) |
| non-Hispanic-Other | 56(5%) | 36(11%) | 92(6%) |
| **Maternal education[a]** | **948(80%)** | **341(95%)** | **1289 (84%)** |
| < High School | 18(2%) | 28(8%) | 46(4%) |
| High School | 89(9%) | 45(13%) | 134(10%) |
| Some college and above | 841(89%) | 268(79%) | 1109 (86%) |
| **Employment** | **1161(99%)** | **306(86%)** | **1467 (96%)** |
| Employed | 922(79%) | 228(75%) | 1150 (78%) |
| Unemployed | 239(21%) | 78(25%) | 317(22%) |
| **Marital Status[b]** | **785(67%)** | **329(92%)** | **1114 (73%)** |
| Married/living with a partner | 680(87%) | 252(77%) | 932(84%) |
| Single | 105(13%) | 77(23%) | 182(16%) |
| **# of people in household** | **1171(99.5%)** | **351(98%)** | **1522 (99%)** |
| 1–3 people | 313(27%) | 117(33%) | 430(28%) |
| 4–6 people | 788(67%) | 213(61%) | 1001 (66%) |
| 7+ people | 70(6%) | 21(6%) | 91(6%) |
| **Current Circumstances[c]** | **1171(99.5%)** | **351(98%)** | **1522 (99%)** |
| Local government ordered to stay at home | 134(11%) | 47(13%) | 181(12%) |
| Local government encouraged to stay at home | 544(46%) | 105(30%) | 649(43%) |
| Local government recommended social distancing | 454(39%) | 172(49%) | 626(41%) |
| No local government recommendation | 39(3%) | 27(8%) | 66(4%) |
| **Household member Household COVID-19 status** | **1169(99%)** | **312(87%)** | **1481 (96%)** |
| Any positive (including respondent) | 85(7%) | 64(21%) | 149(10%) |
| All negative | 1084(93%) | 248(79%) | 1332 (90%) |
| **Total # of acute stress symptoms[d]** | **1128(96%)** | **291(81%)** | **1419 (92%)** |
| 0 symptoms | 135(12%) | 47(16%) | 182(13%) |

*(Continued)*

**Table 1.** (Continued)

| Characteristics, N(%) with data | December 2020—February 2021 | March—May 2021 | Overall |
|---|---|---|---|
| 1–3 symptoms | 412(37%) | 122(42%) | 534(38%) |
| 4–6 symptoms | 417(37%) | 94(32%) | 511(36%) |
| 7–9 symptoms | 164(15%) | 28(10%) | 192(14%) |
| **Cohorts, Location** | | | |
| Brown university Assessment of Myelination and Behavior Across normal Maturation (BAMBAM), Rhode Island | 35(3%) | 76(21%) | 111(7%) |
| Maternal health Influences and Nutrition in Neonatal and Infant dEvelopment (MINNIE), Rhode Island | 28(2%) | 52(15%) | 80(5%) |
| New Hampshire Birth Cohort Study (NHBCS), New Hampshire | 1030(88%) | 0(0%) | 1030 (67%) |
| The Global Alliance to Prevent Prematurity and Stillbirth (GAPPS), Washington | <5(<1%) | <55(<16%) | 57(4%) |
| Columbia Center for Children's Environmental Health (CCCEH) Sibling/Hermanos, New York | <10(<1%) | <25(<7%) | 33(2%) |
| CCCEH Fair Start Study, New York | 36(3%) | 48(13%) | 84(5%) |
| Illinois Kids Development Study (IKIDS), Illinois | 0(0%) | 32(9%) | 32(2%) |
| Chemicals in our Bodies (CIOB), California | 24(2%) | 58(16%) | 82(5%) |

[a]Maternal education data representing the highest level of educational attainment was harmonized into three categories: less than high school, high school, or some college and above.

[b]Marital status was also dichotomized as married/living with a partner or not married (widowed; separated; divorced; single, never married; partnered (boyfriend or girlfriend, not living together).

[c]Current Circumstances was harmonized into four mutually exclusive categories: Local government ordered residents to stay at home; encouraged residents to stay at home; made social distancing recommendations, but did not encourage or order residents to stay at home; and Local government did not make stay at home or social distancing recommendations. If a respondent answered "yes" to multiple situations, then the respondent was put into the most restricted group.

[d]For the 9 stress questions asked in the COVID-19 questionnaire, the total number of questions with score $> = 3$ was considered as the number of symptoms at a significant level ([35]), and was then grouped as 0, 1–3, 4–6, or 7–9 (categorical mid-points were used in multivariate analyses when examining linear trend).

minimal, moderate, and severe pandemic-related traumatic stress symptoms associated with the pandemic.

## Food-related behaviors

We observed that nearly half (49%) of all respondents reported eating more home cooked meals because of the pandemic; only 3% reported eating fewer home cooked meals (**Fig 1**). The proportion eating more home cooked meals was highest among Hispanics compared to other racial/ethnic groups (64% vs. 46%-58%), and among those who tested positive or who lived in a household where someone tested positive for COVID-19 (59% vs. 48%). We also observed that respondents who reported more symptoms of pandemic-related traumatic stress reported eating more home cooked meals (p-trend < 0.001) (**Fig 2**). In multivariable models, Hispanic ethnicity, older age, having fewer people in the household, and reporting more pandemic-related traumatic stress symptoms were all independently associated with the consumption of more home cooked meals (**Table 2**).

One-third (34%) of our respondents reported eating less fast food. Ethnicity was differentially associated with consuming less fast food. A reduction in fast food consumption was most common among Hispanic (51%) vs. Asian (36%), White (32%) and Black (26%) participants. Respondents who reported more symptoms of pandemic-related traumatic stress symptoms were associated with a report of eating less fast food (p-trend < 0.001) (**Fig 2**). In multivariate models, these covariates remained significant independent factors along with older age, which was associated with was consuming less fast food (**Table 2**).

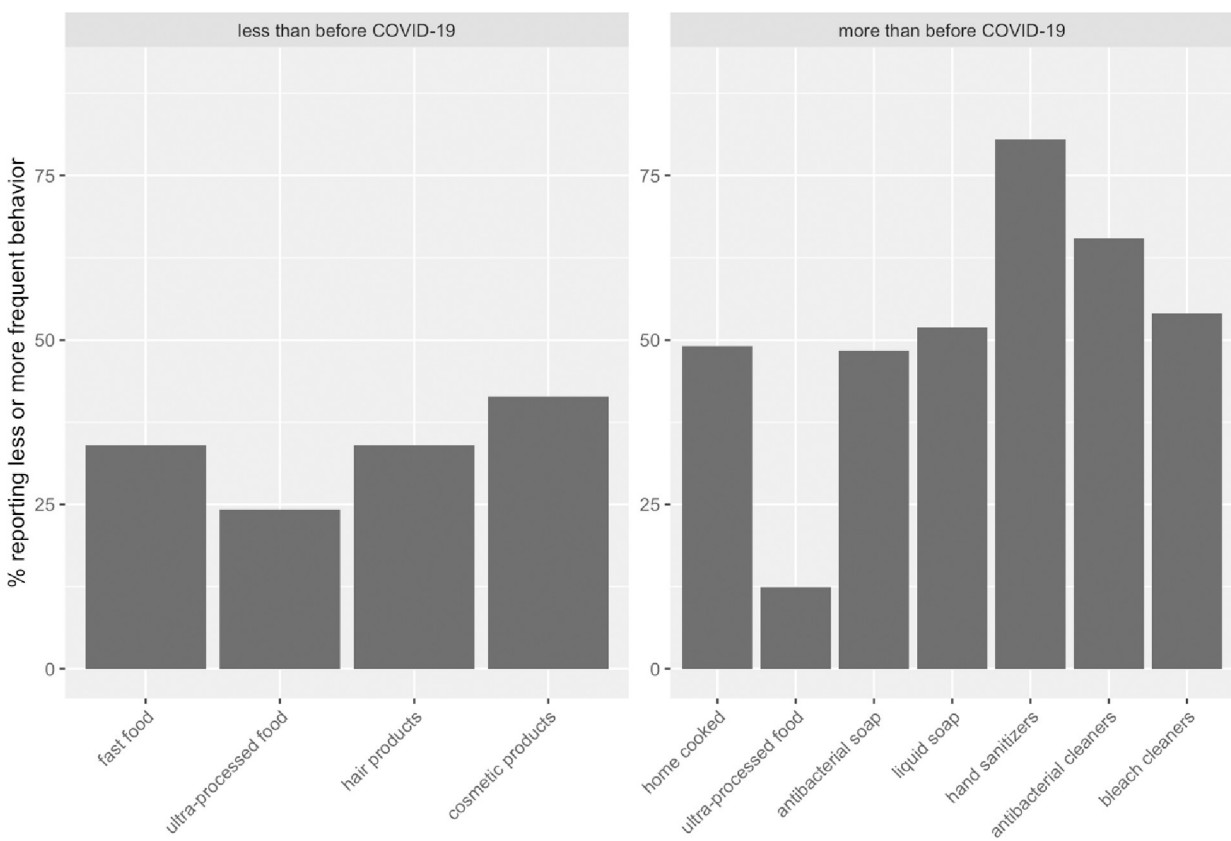

**Fig 1. Exposures reported to be significantly less or more frequent than before the COVID-19 pandemic.**

## Consumption of ultra-processed food

Based on our definition of ultra-processed foods, 12% reported eating more ultra-processed foods and 24% reported eating less processed foods since the start of the pandemic (**Fig 1**).

Being single (19% vs. 11% among partnered, p = 0.01), unemployed (21% vs. 10% among employed, p<0.001) and testing positive or living in a household where someone tested positive for COVID-19 (21% vs. 12% living in a household where all members remained COVID negative, p = 0.001) was associated with more processed food consumption. Ethnicity and race were also associated (p = 0.022) with eating more processed food—Asian (24%), Hispanic (19%), Black (16%) and White (11%) participants. As the number of pandemic-related traumatic stress symptoms increased, the probability of consuming more processed food consumption also increased monotonically (p-trend < 0.001). In multivariable models, unemployment, being single and more pandemic-related traumatic stress symptoms were associated with consumption of more processed foods (**Table 2**).

Older participants were more likely to report eating more processed food, whereas younger participants reported eating less processed food than older respondents (p = 0.01). Being single (33% vs. 22% among partnered, p = 0.002), or Hispanic ethnicity ((55%) vs. non-Hispanic Black (38%), non-Hispanic Asian (24%) and non-Hispanic White (19%), p<0.001) was associated with less processed food consumption. Respondents with less than a high school education reported eating less processed food (57% vs. 39% who completed high school vs. 21% among those who completed at least some college, p<0.001). Those living in households where at least one member tested positive for COVID-19 also reported eating less processed foods (37% vs. 23%, p<0.001). Reporting the highest number of pandemic-related traumatic

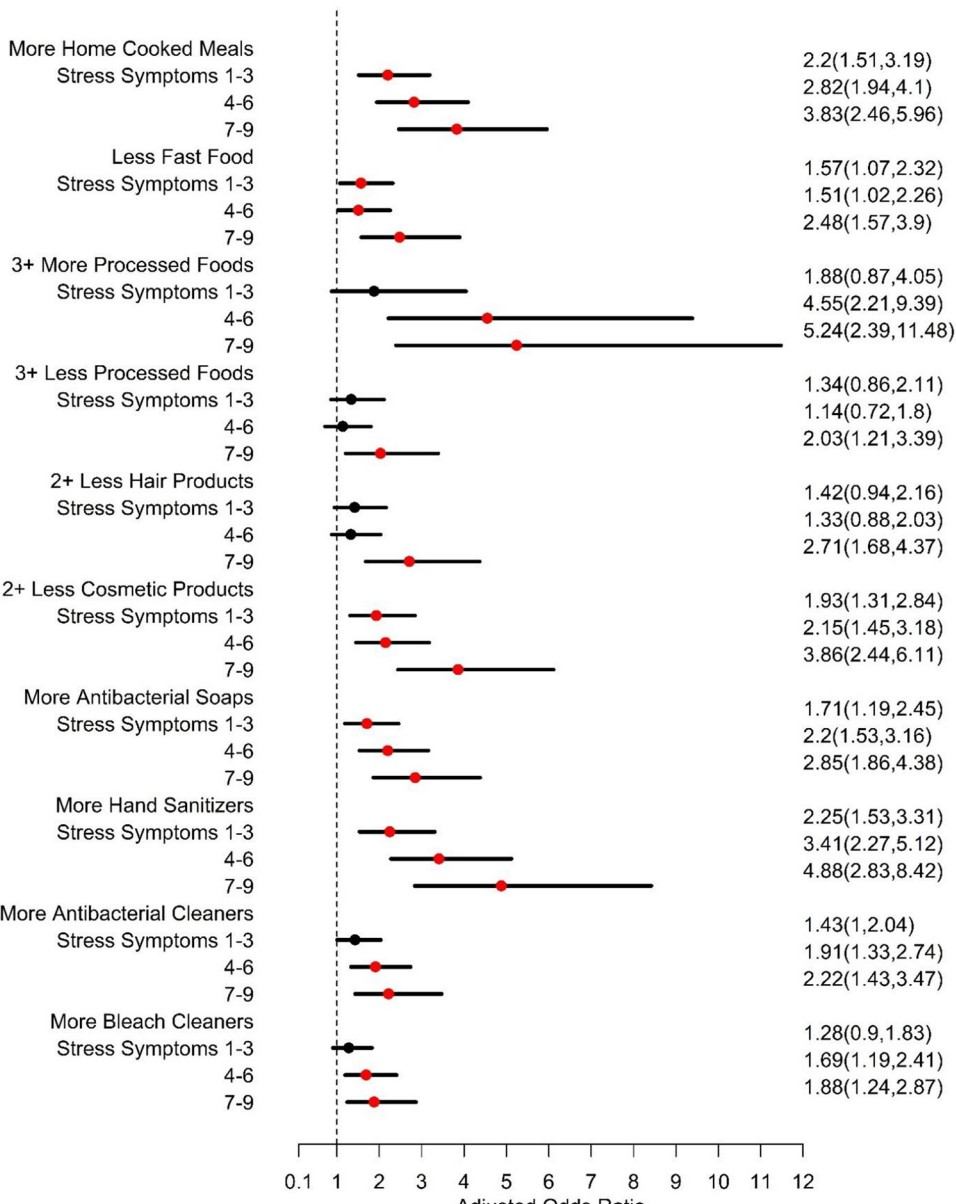

**Fig 2. Adjusted\* odds ratios and 95% confidence intervals\*\* for differences in exposure during vs. before the COVID-19 pandemic by acute stress symptom scores.** \* Adjusted for age, race/ethnicity, education, employment, marital status, number of people in the household, whether any household member tested COVID-19 positive, and whether the cohort was New Hampshire (NH). \*\*Statistically significant results indicated by red marker.

stress symptoms (7–9) were associated with consuming less processed foods (p<0.001). In multivariable models, self-reported Hispanic ethnicity (relative to non-Hispanic White ethnicity), less maternal education, and reporting a high number of pandemic-related traumatic stress symptoms were associated with eating fewer processed foods.

## Use of personal care products

Overall, study participants reported using fewer personal care products, which can be grouped into hair products (including perms or relaxers, hair dye, hair sprays, hair gels) and make-up/

**Table 2. Odds ratios and 95% confidence intervals\* from multivariate models for key exposures and behaviors of interest\*\*.**

| | Less than before COVID-19 | | | | More than before COVID-19 | | | | | |
|---|---|---|---|---|---|---|---|---|---|---|
| | fast food | processed food | hair products | cosmetic products | home cooked meals | processed foods | antibacterial soaps | hand sanitizers | antibacterial cleaners | bleach cleaners |
| **Age of respondent (ref: 19–24 years)** | | | | | | | | | | |
| **25–34 years** | **2.5** | 1.2 | 1.4 | 1.5 | 2.0 | 0.9 | 1.5 | 1.1 | 1.9 | 1.8 |
| | **(1.1, 5.8)** | (0.6, 2.6) | (0.7, 2.9) | (0.7, 3.1) | (0.9, 4.1) | (0.3, 2.4) | (0.7, 3.1) | (0.5, 2.3) | (0.9, 3.9) | (0.9, 3.7) |
| **35–44 years** | **2.4** | 0.9 | 0.8 | 1.1 | 2.1 | 1.0 | 1.7 | 2.0 | **2.5** | 2.1 |
| | **(1.0, 5.5)** | (0.4, 2.0) | (0.4, 1.7) | (0.5, 2.2) | (1.0, 4.3) | (0.4, 2.5) | (0.8, 3.5) | (0.9, 4.2) | **(1.2, 5.3)** | (1.0, 4.4) |
| **45+ years** | **3.0** | 1.1 | 0.8 | 0.9 | **2.7** | 1.1 | 1.5 | 2.1 | **2.5** | 2.1 |
| | **(1.2, 7.4)** | (0.5, 2.7) | (0.4, 1.9) | (0.4, 2.1) | **(1.2, 6.2)** | (0.4, 3.2) | (0.6, 3.3) | (0.8, 5.1) | **(1.1, 5.8)** | (0.9, 4.7) |
| **Race/Ethnicity (ref: non-Hispanic White)** | | | | | | | | | | |
| **Hispanic** | **2.3** | **3.5** | **4.2** | **1.6** | **2.4** | 1.2 | **2.1** | 1.1 | **2.6** | **3.3** |
| | **(1.4, 3.5)** | **(2.2, 5.6)** | **(2.6, 6.6)** | **(1.0, 2.4)** | **(1.5, 3.9)** | (0.7, 2.3) | **(1.3, 3.2)** | (0.6, 1.9) | **(1.6, 4.2)** | **(2.1, 5.3)** |
| **non-Hispanic-Black** | 0.9 | 2.2 | **2.8** | 1.5 | 1.8 | 1.6 | 1.6 | 1.2 | 1.9 | **2.3** |
| | (0.4, 2.0) | (1.0, 5.0) | **(1.3, 6.0)** | (0.7, 3.3) | (0.8, 3.9) | (0.6, 4.5) | (0.7, 3.4) | (0.5, 2.9) | (0.8, 4.5) | **(1.0, 5.1)** |
| **non-Hispanic-Asian** | 1.3 | 1.4 | 1.2 | 1.3 | 1.0 | 2.5 | 1.3 | 0.9 | 0.9 | 0.9 |
| | (0.6, 3.0) | (0.5, 3.6) | (0.5, 3.1) | (0.6, 2.8) | (0.4, 2.2) | (0.9, 6.8) | (0.6, 2.9) | (0.3, 2.5) | (0.4, 2.1) | (0.4, 2.1) |
| **non-Hispanic-Other** | 1.0 | 1.0 | 1.5 | 1.2 | 1.6 | 1.0 | 0.8 | 1.3 | 1.1 | 1.0 |
| | (0.6, 1.6) | (0.6, 1.8) | (0.9, 2.4) | (0.8, 1.9) | (1.0, 2.5) | (0.5, 2.0) | (0.5, 1.2) | (0.7, 2.4) | (0.7, 1.8) | (0.7, 1.6) |
| **Maternal education (ref: some college and above)** | | | | | | | | | | |
| **< High School** | 1.4 | **2.5** | 1.8 | 2.0 | 0.7 | 0.8 | 0.8 | 0.8 | 0.7 | 0.5 |
| | (0.7, 2.8) | **(1.2, 5.2)** | (0.81, 3.9) | (1.0, 4.2) | (0.3, 1.3) | (0.3, 2.0) | (0.4, 1.6) | (0.4, 1.8) | (0.3, 1.4) | (0.3, 1.0) |
| **High School** | 1.3 | **1.8** | 1.2 | 0.9 | 1.0 | 0.7 | 1.1 | **0.6** | 0.8 | 0.8 |
| | (0.8, 2.0) | **(1.1, 3.0)** | (0.7, 1.9) | (0.6, 1.3) | (0.6, 1.5) | (0.4, 1.3) | (0.8, 1.7) | **(0.4, 1.0)** | (0.5, 1.2) | (0.5, 1.2) |
| **Employment (ref: unemployed)** | | | | | | | | | | |
| **employed** | 1.1 | 0.9 | 0.8 | 0.9 | 0.8 | **0.4** | 0.9 | 1.1 | 1.1 | 1.5 |
| | (0.8, 1.4) | (0.6, 1.2) | (0.6, 1.0) | (0.7, 1.1) | (0.6, 1.0) | **(0.3, 0.6)** | (0.7, 1.2) | (0.8, 1.6) | (0.9, 1.5) | (1.2, 2.0) |
| **Marital Status (ref: single)** | | | | | | | | | | |
| **married or living with a partner** | 1.3 | 1.4 | 0.8 | 1.2 | 1.1 | **0.5** | 0.9 | **1.6** | 0.9 | 0.7 |
| | (0.8, 2.0) | (0.9, 2.1) | (0.5, 1.2) | (0.8, 1.8) | (0.8, 1.7) | **(0.3, 0.9)** | (0.6, 1.3) | **(1.0, 2.4)** | (0.6, 1.4) | (0.5, 1.03) |
| **Number of people in household (ref: 1–3 people)** | | | | | | | | | | |
| **4–6 people** | 1.0 | 0.8 | 1.1 | 0.9 | **0.7** | 0.8 | 0.9 | 0.9 | 1.0 | 1.0 |
| | (0.8, 1.3) | (0.6, 1.1) | (0.8, 1.4) | (0.7, 1.1) | **(0.6, 0.9)** | (0.6, 1.2) | (0.7, 1.2) | (0.7, 1.2) | (0.8, 1.2) | (0.8, 1.3) |
| **7+ people** | 1.0 | 0.8 | 1.1 | 0.6 | **0.6** | 1.1 | 1.1 | 0.8 | 0.8 | 1.2 |
| | (0.6, 1.7) | (0.5, 1.5) | (0.6, 1.8) | (0.4, 1.0) | **(0.3, 0.9)** | (0.6, 2.1) | (0.7, 1.7) | (0.4, 1.4) | (0.5, 1.4) | (0.8, 2.0) |
| **Household member COVID-19 status (ref: all negative)** | | | | | | | | | | |

*(Continued)*

**Table 2.** (Continued)

| | Less than before COVID-19 | | | | More than before COVID-19 | | | | | |
|---|---|---|---|---|---|---|---|---|---|---|
| | fast food | processed food | hair products | cosmetic products | home cooked meals | processed foods | antibacterial soaps | hand sanitizers | antibacterial cleaners | bleach cleaners |
| **any positive (incl. respondent)** | 1.1 | 1.2 | 1.1 | 1.2 | 1.4 | 1.6 | 0.8 | 0.7 | 1.1 | 1.2 |
| | (0.7, 1.6) | (0.8, 1.9) | (0.8, 1.7) | (0.9, 1.8) | (1.0, 2.0) | (1.0, 2.6) | (0.5, 1.1) | (0.5, 1.1) | (0.7, 1.6) | (0.8, 1.7) |
| **Cohort group (ref: non-NH cohort)** | | | | | | | | | | |
| **NH cohort** | 1.0 | 0.9 | 0.9 | 0.9 | 1.0 | 0.9 | 1.1 | **1.4** | 1.3 | **1.4** |
| | (0.8, 1.4) | (0.6, 1.2) | (0.6, 1.2) | (0.6, 1.1) | (0.7, 1.3) | (0.6, 1.4) | (0.8, 1.5) | **(1.1, 2.0)** | (1.0, 1.7) | **(1.1, 1.9)** |
| **Total Number of stress symptoms (ref: 0 symptoms)** | | | | | | | | | | |
| **1–3 symptoms** | **1.6** | 1.3 | 1.4 | **1.9** | **2.2** | **1.9** | **1.7** | **2.2** | **1.4** | 1.3 |
| | **(1.1, 2.3)** | (0.9, 2.1) | (1.0, 2.2) | **(1.3, 2.8)** | **(1.5, 3.2)** | (0.9, 4.1) | **(1.2, 2.5)** | **(1.5, 3.3)** | **(1.0, 2.0)** | (0.9, 1.8) |
| **4–6 symptoms** | **1.5** | 1.1 | 1.3 | **2.1** | **2.8** | **4.6** | **2.2** | **3.4** | **1.9** | **1.7** |
| | **(1.0, 2.2)** | (0.7, 1.8) | (0.9, 2.0) | **(1.4, 3.2)** | **(1.9, 4.1)** | **(2.2, 9.5)** | **(1.5, 3.2)** | **(2.3, 5.1)** | **(1.3, 2.8)** | **(1.2, 2.4)** |
| **7–9 symptoms** | **2.5** | **2.0** | **2.8** | **3.9** | **3.9** | **5.3** | **2.9** | **4.8** | **2.2** | **1.9** |
| | **(1.6, 3.9)** | **(1.2, 3.4)** | **(1.7, 4.4)** | **(2.4, 6.1)** | **(2.5, 6.0)** | **(2.4, 11.6)** | **(1.9, 4.4)** | **(2.8, 8.4)** | **(1.4, 3.5)** | **(1.2, 2.9)** |

*statistically significant results (p<0.05) presented in bold font

**respondents had the option to report whether they used more/less/same amount as before the pandemic; here we report behaviors where a significant number reported changing their behavior in either direction.

body products (including nail polish, make-up, perfume, and lotion) (**Fig 1**). Generally, participants who reported using fewer hair products were more likely to be younger (p < 0.001), more likely to be Hispanic (71%), have completed less education (73% less than high school, 50% high school, 30% some college), unemployed (44% vs. 31% employed), single (52% vs. 30% partnered), and lower income (p < 0.001). Households in which a respondent tested positive for COVID-19 were more likely to use fewer hair products (49% vs. 33% for households without a positive test) and participants who experienced more pandemic-related traumatic stress symptoms were more likely to report using fewer hair products (p < 0.001). The trends were similar for reduced cosmetic use. In multivariable models, Hispanic ethnicity, Non-Hispanic Black race and more pandemic-related traumatic stress symptoms were independently associated with using fewer hair products and fewer make-up products (**Table 2**).

Approximately half of all respondents reported using more liquid soaps (52%) and antibacterial soaps (48%) and 81% of respondents reported using more hand sanitizer gels (**Fig 1**). Age was positively associated with using more liquid soap (p < 0.01) and hand sanitizer (p < 0.001) but not antibacterial soap. Racial/ethnicity was associated with antibacterial soap use (e.g., 62% among Hispanics vs. 39–54% in other racial/ethnic groups). More hand sanitizer use was reported among more educated (p < 0.001), employed (p = 0.04), partnered (p < 0.001) and higher income (p < 0.001) respondents. More liquid soap and hand sanitizer gel use was reported among participants who had a household member test positive for COVID-19 (p < 0.01) and the use of all three products was positively associated with pandemic-related traumatic stress symptoms (p < 0.001). In multivariable models, only pandemic-related traumatic stress symptoms were consistently associated with increased use of liquid soap, antibacterial soap, and hand sanitizer gels (p < 0.001).

## Use of consumer products

Two-thirds of respondents reported using more antibacterial cleaners and 54% reported using more bleach-containing cleaning products since the start of the pandemic. Older respondents were more likely to use more antimicrobial cleaning products (p < 0.01) and bleach-containing products (p = 0.02) and those who self-identify as Hispanic were more likely to use bleach-containing products and antibacterial cleaners relative to other race/ethnic groups (p < 0.001 and p = 0.04, respectively). Those who were employed (56% vs. 47%) and those who were single (60% vs. 51%) reported using more bleach containing products. Participants who reported experiencing more pandemic-related traumatic stress were also more likely to use more antimicrobial and bleach-containing products (p < 0.001). In multivariable models, Hispanic ethnicity and pandemic-related stress predicted more use of antimicrobial and bleach-containing cleansers. For antimicrobial cleaners, increasing age was an independently significant predictor and for bleach-containing cleaners, Non-Hispanic Blacks and those from the NH-based cohort also reported increased use (**Table 2**).

## Discussion

### Key findings

We identified some changes in behaviors that are associated with chemical exposures during the COVID-19 pandemic in expected directions. For example, since the start of the pandemic, we found that overall, respondents reported using fewer personal care products like hair dyes and make-up but more sanitizing and cleaning products.

While we did not measure actual exposure change here, based on the existing literature, we may expect that these behavior changes reflect changes in environmental chemical exposures. We can infer that some behaviors might lead to less chemical exposure (e.g., less consumption of fast foods and less use of personal care products may be associated with lower exposure to some phthalates and phenols) [27, 45] or more chemical exposure (e.g., more use of personal and household cleansers may be associated with higher exposure to quaternary ammonium compounds and glycol ethers) [46]. Ultra-processed food consumption has been associated with increased exposure to chemical classes including phthalates and phenols [47].

Multivariable analyses indicated that there were notable demographic and geographic trends associated with increased or decreased behaviors potentially associated with chemical exposures. For example, respondents who self-identified as being of Hispanic ethnicity reported many behavior changes, and increasing age was associated with less fast food, but more antibacterial cleanser use. Despite small sample sizes in some strata, this information may be used to anticipate subpopulations who are more likely to be exposed to a variety of chemicals associated with these behavior changes. These results can be further explored and potentially validated with biomarker studies to determine whether behavior changes observed here associate with chemical exposure levels. Therefore, we may be able to anticipate subpopulations who may be at higher risk for health outcomes associated with these exposures.

Overall, we found that symptoms of pandemic-related traumatic stress were consistently associated with the pandemic-related behavior changes, indicating that those who experienced the most significant shifts in behavior (and likely environmental exposures) also reported more pandemic-related traumatic stress symptoms. However, we cannot discern from our study whether stress leads to behavior change, whether behavior change leads to stress or whether they occur concurrently. There is a growing literature indicating that chemical and psychosocial stressors interact, exacerbating health outcomes associated with either one independently[44]. Therefore, the findings reported here highlight the importance of considering

pandemic-related stress when examining the impact of exposures on health outcomes (and vice versa). We also note that because both metrics were reported by the same individual, it is possible that single reporter bias may have influenced the likelihood of these associations. Finally, while our survey questions specifically focused on symptoms and experiences since the start of the COVID-19 pandemic, it is not possible to rule out the possibility that other circumstances outside of the pandemic itself influenced the experience and reporting of these symptoms.

From our initial survey, it is not clear whether behavior changes will be sustained as the COVID-19 pandemic evolves. The administration of our questionnaire spanned December 2020 to May 2021 and continues to be administered to many of the cohorts across the consortium. It will be important to continue to monitor pandemic related behavior change as the pandemic evolves and waves of pandemic severity wax and wane.

## Strengths

To our knowledge, this study is the first of its kind to evaluate how the COVID-19 pandemic may have influenced behaviors associated with chemical exposures and the first to look at how these behaviors change with pandemic-related stress. Because this investigation was conducted within the ECHO-wide cohort, we were able to take advantage of a relatively large sample size of women that was somewhat diverse in terms of geography, race, and ethnicity, and socioeconomic status allowing us to detect patterns in behaviors and to make inferences based on previous literature about pandemic-related effects of these behaviors on environmental exposures within subgroups (specific population strata).

## Limitations

The CEE survey enabled us to capture dynamic behaviors that may lead to shifts in environmental exposure during critical time windows. We acknowledge that in some cases, self-reported behaviors may not correlate with associated environmental exposures, there are some environmental chemicals that cannot be well assessed via questionnaire, and retrospective survey responses may be subject to differential recall. Further, our goal was to create a relatively short questionnaire that could be added without increasing participant burden. We, therefore, did not assess baseline behaviors but rather captured behavior change. There are some limitations with respect to the population to whom our survey was administered. Many of our respondents were from one NH-based cohort in a rural setting. The way that the questionnaire was rolled out for administration in the participating cohorts, including the timing of IRB approval and whether the cohorts could "blast" the questionnaires to all participants at once or whether they administer them one-by-one, also contributed to the variation in response across the cohorts. In addition, our ECHO sample did not include enough pregnant women to examine this subgroup separately and changes in behaviors (and related exposures) during pregnancy are important to understand as they have direct implications on many developmental processes. However, pregnant women may change their behavior due to pregnancy rather than due solely to the pandemic; therefore, a larger sample size would be needed to understand these shifts. Finally, cohorts offered this survey according to their local cohort protocols. Therefore, not every member of every cohort included was offered a survey during the study period, precluding our ability to calculate a survey response rate.

## Conclusions

Among a subset of participants from a national cohort study, we characterized changes in behaviors related to environmental chemical exposures during the COVID-19 pandemic. As

part of the ECHO consortium, by repeating this analysis using a panel of urinary biomarkers reflecting chemical exposure, we can confirm whether the trends in pandemic-related behavior change reported here do, in fact, result in shifts in biomarkers of internal dose. Changes in certain behaviors differed by sociodemographic factors (e.g., Hispanic ethnicity) and nearly all pandemic-related changes in behaviors we assessed were related to more reported pandemic-related traumatic stress symptoms. These findings highlight that behavior change associated with the pandemic and experiencing pandemic-related traumatic stress often co-occurred. Thus, interventions and campaigns targeting the reduction of environmental exposures, pandemic-related traumatic stress as well as those that facilitate behavior change may lead to the largest health benefits in the context of a pandemic. Future biomarker analyses can determine whether behavior changes observed here associate with chemical exposure levels.

## Supporting information

**S1 File. ECHO COVID-19 changes in environmental exposures questionnaire (version 2, April 2021).**
(PDF)

## Acknowledgments

The authors wish to thank our ECHO colleagues, the medical, nursing and program staff, as well as the children and families participating in the ECHO cohorts. We also acknowledge the contribution of the following ECHO program collaborators: Coordinating Center: *Duke Clinical Research Institute*, *Durham*, *North Carolina*: Smith PB, Newby KL, Benjamin DK; Data Analysis Center: *Johns Hopkins University Bloomberg School of Public Health*, *Baltimore*, *Maryland*: Jacobson LP; *Research Triangle Institute*, *Durham*, *North Carolina*: Parker CB. The content is solely the responsibility of the authors and does not necessarily represent the official views of the National Institutes of Health.

## Author Contributions

**Conceptualization:** Megan E. Romano, Xiuhong Li, Lisa P. Jacobson, Amy E. Margolis, Deborah H. Bennett, Jessie P. Buckley, Leonardo Trasande, Tracey J. Woodruff, Frederica P. Perera, Margaret R. Karagas.

**Data curation:** Julie B. Herbstman, Lisa P. Jacobson, Ghassan B. Hamra, Jessie P. Buckley, Trina Colburn, Sean Deoni, Lori A. Hoepner, Rachel Morello-Frosch, Kylie Wheelock Riley, Sheela Sathyanarayana, Susan L. Schantz, Frederica P. Perera, Margaret R. Karagas.

**Formal analysis:** Julie B. Herbstman, Megan E. Romano, Xiuhong Li, Ghassan B. Hamra.

**Funding acquisition:** Julie B. Herbstman, Lisa P. Jacobson, Deborah H. Bennett, Sean Deoni, Sheela Sathyanarayana, Susan L. Schantz, Leonardo Trasande, Tracey J. Woodruff, Frederica P. Perera, Margaret R. Karagas.

**Investigation:** Julie B. Herbstman.

**Methodology:** Julie B. Herbstman, Amy E. Margolis.

**Project administration:** Julie B. Herbstman.

**Resources:** Julie B. Herbstman.

**Supervision:** Julie B. Herbstman.

**Visualization:** Julie B. Herbstman, Joseph M. Braun.

**Writing – original draft:** Julie B. Herbstman, Megan E. Romano, Xiuhong Li, Lisa P. Jacobson, Amy E. Margolis, Ghassan B. Hamra, Deborah H. Bennett, Joseph M. Braun, Jessie P. Buckley, Trina Colburn, Sean Deoni, Lori A. Hoepner, Rachel Morello-Frosch, Kylie Wheelock Riley, Sheela Sathyanarayana, Susan L. Schantz, Leonardo Trasande, Tracey J. Woodruff, Frederica P. Perera, Margaret R. Karagas.

**Writing – review & editing:** Julie B. Herbstman, Megan E. Romano, Xiuhong Li, Lisa P. Jacobson, Amy E. Margolis, Ghassan B. Hamra, Deborah H. Bennett, Joseph M. Braun, Jessie P. Buckley, Trina Colburn, Sean Deoni, Lori A. Hoepner, Rachel Morello-Frosch, Kylie Wheelock Riley, Sheela Sathyanarayana, Susan L. Schantz, Leonardo Trasande, Tracey J. Woodruff, Frederica P. Perera, Margaret R. Karagas.

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
