## [Decision Letter · Decision Letter 0]

22 Jun 2022

PONE-D-22-05387Characterizing changes in behaviors associated with chemical exposures during the COVID-19 pandemicPLOS ONE

Dear Dr. Herbstman,

Thank you for submitting your manuscript to PLOS ONE. After careful consideration, we feel that it has merit but does not fully meet PLOS ONE’s publication criteria as it currently stands. Therefore, we invite you to submit a revised version of the manuscript that addresses the points raised during the review process.

Please see the comments and specific suggestions for improving the manuscript from the reviewers below.

We look forward to receiving your revised manuscript.

Kind regards,

Aaron Specht

Academic Editor

PLOS ONE

Journal Requirements:

3. Please update your submission to use the PLOS LaTeX template. The template and more information on our requirements for LaTeX submissions can be found at http://journals.plos.org/plosone/s/latex

" Research reported in this publication was supported by the Environmental influences on Child Health Outcomes (ECHO) program, Office of The Director,

National Institutes of Health, under Award Numbers U2COD023375 (Coordinating Center), U24OD023382 (Data Analysis Center), U24OD023319 (PRO Core), and UH3OD023290 (Columbia University, New York, New York: Perera FP, Herbstman JB); UH3OD023275 (Dartmouth College, Hanover, New Hampshire: Karagas MR); UH3OD023272 (University of Illinois, Urbana: Schantz SL, University of California, San Francisco: Woodruff T, University of California, Berkeley, Morello-Frosch, R); UH3OD023271 (University of Washington, Seattle: Karr C, Sathyanarayana S); and UH3OD023313 (Memorial Hospital of Rhode Island,Pawtucket: Deoni S, D’Sa VA; Brown University, Providence,RI: Braun J)"

" Research reported in this publication was supported by the Environmental influences on Child Health Outcomes (ECHO) program, Office of The Director, National Institutes of Health, under Award Numbers U2COD023375 (Coordinating Center), U24OD023382 (Data Analysis Center), U24OD023319 (PRO Core), and UH3OD023290 (Columbia University, New York, New York: Perera FP, Herbstman JB); UH3OD023275 (Dartmouth College, Hanover, New Hampshire: Karagas MR); UH3OD023272 (University of Illinois, Urbana: Schantz SL, University of California, San Francisco: Woodruff T, University of California, Berkeley, Morello-Frosch, R); UH3OD023271 (University of Washington, Seattle: Karr C, Sathyanarayana S); and UH3OD023313 (Memorial Hospital of Rhode Island, Pawtucket: Deoni S, D’Sa VA; Brown University, Providence,RI: Braun J)."

Reviewers' comments:

Reviewer's Responses to Questions

**Comments to the Author**

1. Is the manuscript technically sound, and do the data support the conclusions?

Reviewer #1: Yes

Reviewer #2: Partly

Reviewer #3: Partly

2. Has the statistical analysis been performed appropriately and rigorously? 

Reviewer #1: Yes

Reviewer #2: Yes

Reviewer #3: Yes

3. Have the authors made all data underlying the findings in their manuscript fully available?

Reviewer #1: Yes

Reviewer #2: No

Reviewer #3: No

4. Is the manuscript presented in an intelligible fashion and written in standard English?

Reviewer #1: Yes

Reviewer #2: Yes

Reviewer #3: Yes

5. Review Comments to the Author

Reviewer #1: This cross-sectional survey of behavior-change related to environmental exposure among 1535 members of the ECHO-wide Cohort identified patterns of behavior change among cohort members during the COVID-19 pandemic that could have altered their environmental exposures, particularly through altered diet and the use of cleaners and personal grooming products. This study examines a novel research question related to the COVID-19 pandemic. It is exhaustive in terms of assessing a wide range of potential behavior changes and testing all potential predictors of behavior. It is too bad that it does not yet have information on how much behaviors changed (A lot? A little? Enough to make a difference to health-relevant exposures?) or, as the author’s point out, any direct assessment of chemical exposures. But there are plans to gather such information. In its current form this manuscript provides preliminary evidence of a potentially important trend in environmental exposures among the US population over the past 2 years, one which could motivate further studies / be of use to researchers interested in environmental exposures who might hope to leverage the natural experiment of the pandemic. I note a number of issues that, if addressed, could improve the manuscript.

Major comment

A key limitation of this study is that its wider utility likely hinges on its potential to generalize to the larger US. However 67% of respondents were from New Hampshire and the respondents were largely white, well-educated, and free of covid infection. What I believe is currently missing is a breakdown of what percent of respondents were from each of the 9 cohorts / states (this could be added to Table 1) and some discussion of to what extent we can generalize these findings beyond this particular sample of participants. The potential for wide geographic coverage in this study could be a significant strength - but is this sample generally representative of any populations?

Relatedly, I struggled at times as a reader to identify the clear utility of this study in its current form for the wider field (and not just the ECHO-wide Cohort researchers who plan to follow-up with a panel of urinary biomarkers). Why not just wait for the biomarkers to create a more complete research product? I suspect that the goal is to provide evidence of a potential national trend in behavior change so that other researchers need not wait for the next level of evidence before investigating further. But this was not stated explicitly. I say this because I think that the authors could spend more time throughout the manuscript better enumerating the motivations behind the study and the ultimate implications (in the abstract and introduction in particular). The idea that “prevention strategies and campaigns that limit environmental exposures, support stress reduction, and facilitate behavioral change may lead to the largest health benefits in the context of a pandemic” struck me as speculative and overly broad / non-specific to this particular manuscript’s findings, which are that participants reported changing their behavior during the pandemic, with participants reporting more stress also more likely to report behavior change. To me the big selling point is that this study identifies the exposures for which the pandemic may likely be a useful natural experiment - but perhaps there are other implications I am missing as well.

Minor comments

Minor comments are provided on lines or sections below. A number of typos throughout hindered readability.

Abstract

- Rationale for the study was not clearly articulated before the methods were described. (e.g., Objectives).

- Line 21 - ECHO-wide Cohort is mentioned but not described. A few brief words here would aid the reader unfamiliar with ECHO and provide necessary context for the study.

- Line 30 - “increased COVID-related stress” suggests longitudinal change over time. Suggest replacing with “greater COVID-related stress” unless the comparison was dichotomous (stress vs. no stress), in which case the language should reflect that instead.

- Line 32 - suggest stating “less chemical exposure”

Line 83 - I believe the causal evidence is overstated. Suggest changing text to “psychosocial stress may interact…”

Line 92. Please describe the ECHO-wide Cohort.

Line 94 - typo “?”

Line 127 - the manual acronym is DSM-5, not DSM-V. Suggest spelling out in the first instance.

Line 129 - Further context on the DSM-5 definition of traumatic events may be useful here, as the definition is narrow and it is possible that many of the respondents may not have been truly exposed to a traumatic event (particularly given the low COVID-infection rates reported). The DSM-5 definition of a traumatic event requires “actual or threatened death, serious injury, or sexual violence.”

Line 172 - what is NOVA?

Line 401 - typo “that reduce”

Table 2

- I think it is confusing to place “processed food” in both the “less than” and “more than” columns. I suggest picking the question that is of most interest and dropping the other use. Same comment for Figure 1.

- Table 2 also makes clear the large number of tests conducted. I appreciate that these predictors of behavior change tests are not the primary study tests, however the readers may be more confident in the findings if the authors note which associations remain significant after some form of correction for multiple testing (e.g., FDR).

Figure 1 - Figure 1’s title may mislead the casual reader into concluding that the y-axis describes how much a particular behavior has increased or declined on average for respondents when, as I understand it, the figure instead displays the the percent of respondents who reported doing less or more of something. I suggest amending the title or adding a footnote that makes it clear the “percent” on the y-axis is the “percent of respondents reporting less frequent or more frequent behavior”.

Figure 2 makes a clear and compelling case for behavior change increasing alongside stress symptoms. Well done. I suggest moving the effect estimates on the right side so that they overlap less with the confidence intervals for the 3+ More Processed Food category.

Authors note that they “cannot discern from our study whether stress leads to behavior change, whether

behavior change leads to stress or whether they occurred concurrently.” The authors may also wish to consider that having the same individual rate their stress and their behavior change may result in inflated estimates of an association owing to single-method, single-reporter bias (e.g., owing to participant reporting style, personality, etc.). Having third party or objective measures of one of these variables may give better estimates of the true association.

Reviewer #2: This is an interesting analysis on an important topic; and there is a lot of interesting data that were collected and analyzed. At the same time, there is concern regarding the way the study and results are presented: the manuscript is highly focused on the fact that the behavior changes during the pandemic likely relate to changes in environmental exposure and health risk - most of the text in the abstract, introduction, and discussion are focused on this point. However, no data were collected directly about environmental exposures, which means that the major talking points of the manuscript are speculative, rather than based on data presented. The data collected include whether behaviors increased/decreased and stress. This is still important and interesting; foccusing more on the data/results that were observed in this specific study is recommended. Additional comments are below.

*Abstract: Overall, the abstract seems to be fairly general. The amount of background could be shortened. The main finding here is that increased stress is correlated with more behavior change; but the majority of the abstract focuses on something that wasn’t measured – the impact of this on environmental exposures. In this light the conclusions of the abstract do not seem to directly derive from the results of this study.

*Abstract. Key specific, numeric, results should be included.

*Line 29: What is meant by “common predictors”?

* Line 121: The authors state that data collection is ongoing; what is the rationale for publishing the results now before the remainder of the data are collected?

*Methods/CEE Survey: Was this survey pilot tested or otherwise assessed for quality?

*Methods/Acute Stress Survey: It appears that the authors have already published about development of the acute stress survey; however, it would be good to explicitly state whether these adaptations were validated in the text here as well.

*Line 153-4: It’s very reasonable to think behaviors changed as the pandemic progressed; but what was the reason to make a comparison based on just early vs. late and was there a reason for selection of March 1, 2021 as the cutoff date?

*Line158-9: “Our analyses focused on outcomes that varied” It’s not clear what is intended by this statement. Also, in this section it seems that you only selected outcomes where none of the participants responded that they had no change. Although I see the explanation further down in the paragraph, this section is confusing and could be clarified.

* Line 197: the percent missing for marital status is fairly high; would this affect the accuracy of the imputation?

* Results: How do the demographics of those who are included in this analysis correspond to the overall ECHO cohort?

* Line 367: I wouldn’t refer to the fact that you are “speculating” about changes in environmental exposure as a strength of the study.

* Line 389: I don’t understand why a survey response rate couldn’t be calculated based on the number of people offered the survey and the number who completed it.

* Line 392: what is meant by a “virtual” cohort study? Also, can this really be called national when the responses are dominated by rural women from NH?

Reviewer #3: Line 39-42: Recent research indicates that air pollution has returned to pre-pandemic levels as recently as February 2021 – please adjust to reflect these data or change the wording to reflect that they have rebounded.

Line 121-125: The study is primarily coming from New Hampshire, which is a very racially homogenous (93% are white alone) state and low in overall population, comparatively speaking. This is an impressive study – as such, what is your rationale for reporting premature results from mostly NH alone, given the substantial generalizability issues and potential for bias? I worry that in trying to tease out racial effects, you’re instead looking at a very small group that could be experiencing heightened stress already because of being an extreme minority, as one example.

Line 147-155: Are you saying here that you are comparing impacts from early COVID outbreaks to later stage COVID (e.g., post vaccine)? If so, it may benefit you to explicitly mention this, as there are many disparate factors that can impact these results.

Line 181-185: It seems like stress and eating behaviors are a bidirectional relationship. Do you have some method to help control for this?

Line 211-214: This is a small percentage of participants who are pregnant. My impression was that this was an area of interest, as pre and perinatal outcomes are highlighted as very important in the introduction.

Line 209-223: I am reminded of the many other stressors that were extremely likely co-occur during this same time in these states; namely, there has been a good deal of research identifying that political stress is matched alongside pandemic stress, with pandemic stress decreasing slightly over time while political stress remained static. There are major differences here by race, ethnicity, and occupation. Do you have any way of controlling for other stressors? Do the COVID questions specifically inquire about stress related to the outbreak of COVID?

Line 255-265: Are these significant? If so, please specify.

Line 267-278: Please specify which relationships are not significant.

Line 276-278: Fascinating.

Line 283-287: These factors seem like they would be highly correlated. Is there a risk of collinearity here?

Line 280-293: I am confused by the decision to use percentages and p values separately and struggle to track what relationships are significant and to what magnitude, both in size of difference and significance. This undermines the strength of the paper.

Line 330-335: I am confused by this paragraph. Please try and clarify the message.

Line 337-344: Could this not be because of the sample sizes? I don’t think it’s fair to assume that these results can be used to anticipate subpopulations who are more likely to be exposed to a variety of chemicals, especially in the context of the complaints I have raised above.

Line 346-354: This is a great paragraph.

Line 362-368: While I do agree with most of this, this is not a diverse population in terms of geography, race, or ethnicity. This statement should be removed.

Line 370-389: I appreciate the candor by which the authors reflect on the limitations of the study. However, I feel it is a major flaw that you cannot truly account for COVID-19 related stress alone. My understanding of the paper is that you are assessing for stress during COVID-19, but not directly measuring stress as a result of COVID-19 alone. Is this a fair assessment? If so, I think this should be discussed in this section.

Line 391-403: I think the authors next step should be to analyze this same relationship in a complete population, rather than the limited scope identified in this paper.

6. PLOS authors have the option to publish the peer review history of their article (what does this mean?). If published, this will include your full peer review and any attached files.

Reviewer #1: No

Reviewer #2: No

Reviewer #3: No

---

## [Author Response · Author response to Decision Letter 0]

18 Sep 2022

We have included this as an uploaded document with better formatting; also pasted here. 

We thank the three reviewers and the editors for their careful read and constructive comments. Please find a detailed point-by-point response below. 

Reviewer 1 (R1) noted that this is addressing a novel research question related to the COVID-19 pandemic. Responses to specific concerns are listed below (first 3 are Major Comments, remainder are Minor Comments):

1. The reviewer laments that it is too bad that we could not evaluate how much behavior changed as a function of the pandemic (e.g., a little? a lot?) and that we do not have a direct measure of exposure. (Major Comment)

 Response: we agree. We needed to keep the questionnaire as short as possible to minimize participant burden (particularly during the pandemic). Therefore, we were not able to obtain data as granular as we may have wanted. In a subset of participants, we will have direct measures of exposure (data forthcoming)—see #3 below; however, we feel that the questionnaire data can provide novel insight into how behaviors may have changed as a function of the pandemic, which is the focus of this analysis. 

2. Reviewer commented on our inability to generalize our results to the larger U.S. population and describe how the study would benefit from a breakdown of the proportion of respondents were from each of the 9 cohorts in Table 1. (Major Comment) 

 Response: we agree that within this paper (and within the ECHO consortium cohort, more generally), the study population does not reflect the general U.S. population. We added this information to Table 1. We have also added text about the generalizability/external validity of this study sample (lines 94-100). 

3. The reviewer suggested that we wait until we have the biomarker data available to publish along with the questionnaire data. They also suggest that we enumerate the motivations behind the study more clearly in the abstract and introduction. (Major Comment)

 Response: We will only have biomarker data on a subset of individuals in this study. Therefore, inferences about changes in exposure as a result of behavior change may be stronger when using a surrogate measure of exposure (e.g., self-report) where we have a more representative sample. In addition, it is important to note that behaviors studied in this paper (rather than the biomarker levels themselves) can be the targets of intervention. We have added text to the Abstract (lines 37-39), Introduction (lines 105-106), Discussion (lines 375-377) and Conclusion (lines 442-444) to more clearly state our objective and conclusions. 

4. Rationale for the study was not clearly articulated before the methods were described. (e.g., Objectives). 

 Response: we have provided a rationale for the research in both the Abstract (line 23) and in the Introduction (line 102). 

5. Line 21 - ECHO-wide Cohort is mentioned but not described. A few brief words here would aid the reader unfamiliar with ECHO and provide necessary context for the study. Line 92. Please describe the ECHO-wide Cohort.

 Response: text was added on lines 21-23 and 94-100 to provide more context. 

6. Line 30 - “increased COVID-related stress” suggests longitudinal change over time. Suggest replacing with “greater COVID-related stress” unless the comparison was dichotomous (stress vs. no stress), in which case the language should reflect that instead.

 Response: we agree; suggested text was substituted

7. Line 32 - suggest stating “less chemical exposure”

 Response: suggested text was substituted.

8. Line 83 - I believe the causal evidence is overstated. Suggest changing text to “psychosocial stress may interact…”

 Response: suggested text was substituted.

9. Line 94 - typo “?”

 Response: it is not clear what “?” the reviewer is referring to; therefore, no changes were made. 

10. Line 127 - the manual acronym is DSM-5, not DSM-V. Suggest spelling out in the first instance.

 Response: suggested text was substituted.

11. Line 129 - Further context on the DSM-5 definition of traumatic events may be useful here, as the definition is narrow and it is possible that many of the respondents may not have been truly exposed to a traumatic event (particularly given the low COVID-infection rates reported). The DSM-5 definition of a traumatic event requires “actual or threatened death, serious injury, or sexual violence.”

 Response: We have added text to the section labelled “Pandemic-related Traumatic Stress (PTS) symptoms measured in the COVID-19 Survey (beginning on line 135). Within this section, we provide more details about how we define pandemic-related stress and the rationale for this measure. 

12. Line 172 - what is NOVA?

 Response: NOVA is validated classification system; it is not an abbreviation and a reference was provided. A more detailed description was added (page 195-197) but readers are encouraged to review the reference which provides more detail than space allows here. 

13. Line 401 - typo “that reduce”

 Response: typo corrected

14. Table 2: I think it is confusing to place “processed food” in both the “less than” and “more than” columns. I suggest picking the question that is of most interest and dropping the other use. Same comment for Figure 1.

 Response: Because our hypothesis is bi-directional (e.g., we hypothesize that the pandemic may in some circumstances increase processed food consumption and, in some contexts, reduce processed food consumption), we have elected to keep both categories in both Table 2 and Figure 1. 

15. Table 2 also makes clear the large number of tests conducted. I appreciate that these predictors of behavior change tests are not the primary study tests, however the readers may be more confident in the findings if the authors note which associations remain significant after some form of correction for multiple testing (e.g., FDR).

 Response: Thank you for this comment. While we agree that we conducted many tests, because there are no primary questions or hypotheses (as the reviewer points out), adjusting testing thresholds for multiple comparisons is not appropriate (Rothman et al. Epidemiology, v1, n1, 1990).

16. Figure 1 - Figure 1’s title may mislead the casual reader into concluding that the y-axis describes how much a particular behavior has increased or declined on average for respondents when, as I understand it, the figure instead displays the percent of respondents who reported doing less or more of something. I suggest amending the title or adding a footnote that makes it clear the “percent” on the y-axis is the “percent of respondents reporting less frequent or more frequent behavior”.

 Response: Changed as suggested

17. Figure 2 makes a clear and compelling case for behavior change increasing alongside stress symptoms. Well done. I suggest moving the effect estimates on the right side so that they overlap less with the confidence intervals for the 3+ More Processed Food category.

 Response: Changed as suggested

18. Authors note that they “cannot discern from our study whether stress leads to behavior change, whether behavior change leads to stress or whether they occurred concurrently.” The authors may also wish to consider that having the same individual rate their stress and their behavior change may result in inflated estimates of an association owing to single-method, single-reporter bias (e.g., owing to participant reporting style, personality, etc.). Having third party or objective measures of one of these variables may give better estimates of the true association.

 Response: Additional possibility noted (line 388-390). 

Reviewer #2 (R2) notes that “this is an interesting analysis on an important topic; and there is a lot of interesting data that were collected and analyzed”. They also point out a number of concerns about the way the study and results are presented. 

19. The manuscript is highly focused on the fact that the behavior changes during the pandemic likely relate to changes in environmental exposure and health risk - most of the text in the abstract, introduction, and discussion are focused on this point. However, no data were collected directly about environmental exposures, which means that the major talking points of the manuscript are speculative, rather than based on data presented.

 Response: We acknowledge throughout the paper that we have not measured actual exposures but rather we have assessed behaviors that have previously been shown to be associated with exposure. Where possible, we have provided references to support our inferences based on the data collected and presented. 

20. Abstract: Overall, the abstract seems to be fairly general. The amount of background could be shortened. The main finding here is that increased stress is correlated with more behavior change; but the majority of the abstract focuses on something that wasn’t measured – the impact of this on environmental exposures. In this light the conclusions of the abstract do not seem to directly derive from the results of this study. Key specific, numeric, results should be included.

 Response: Abstract has been edited accordingly to remove the inferences that were not directly measured.

21. Line 29: What is meant by “common predictors”?

 Response: “Common” was changed to “most frequent”

22. Line 121: The authors state that data collection is ongoing; what is the rationale for publishing the results now before the remainder of the data are collected? 

 Response: As there is no end date for the pandemic, we are continuing to track behaviors that may change as a result of the pandemic. We expect these behaviors to be time-varying, as the nature/scale/scope of the pandemic continues to evolve. As there is no natural stopping point, we elected to publish the information we have now. 

23. Methods/CEE Survey: Was this survey pilot tested or otherwise assessed for quality?

 Response: The survey was assessed by multiple components of the ECHO consortium, including a team of cohort investigators representing cohorts across the country and the data coordinating center. In an effort to get the questionnaire into the field as quickly as possible in response to a pandemic with changing scale/scope/magnitude, we did not formally pilot test it among study participants. 

24. Methods/Acute Stress Survey: It appears that the authors have already published about development of the acute stress survey; however, it would be good to explicitly state whether these adaptations were validated in the text here as well.

 Response: While we currently have a paper under review that describe the development of our pandemic-related traumatic stress scale, we feel that the information we provide in the text (section beginning on line 135) as well as a reference to the psycharchives pre-print provides adequate information to interpret the data presented. 

25. Line 153-4: It’s very reasonable to think behaviors changed as the pandemic progressed; but what was the reason to make a comparison based on just early vs. late and was there a reason for selection of March 1, 2021 as the cutoff date?

 Response: This cut-point of 3/1/2021 was somewhat arbitrary; however, it did represent 1 year into the pandemic. We have added this language to the text to clarify (lines 176-177).

26. Line158-9: “Our analyses focused on outcomes that varied” It’s not clear what is intended by this statement. Also, in this section it seems that you only selected outcomes where none of the participants responded that they had no change. Although I see the explanation further down in the paragraph, this section is confusing and could be clarified.

 Response: Clarifying text was added/moved (lines 179-182).

27. Line 197: the percent missing for marital status is fairly high; would this affect the accuracy of the imputation?

 Response: We do not think so. For missing at random data, multiple imputation reduces bias even when the percent of missingness is large [Reference paper: The proportion of missing data should not be used to guide decisions on multiple imputation. Paul Madley-Dowd, Rachael Hughes, Kate Tilling, Jon Heron. J Clin Epidemiol. 2019 Jun;110:63-73.Epub 2019 Mar 13. PMID: 30878639 PMCID: PMC6547017])

28. Results: How do the demographics of those who are included in this analysis correspond to the overall ECHO cohort?

 Response: It is difficult to assess how the participants included in this study are the same/different as those in the whole consortium. The consortium is dynamic (still recruiting) and sampling for this survey, as we describe, was not random (line 127-134). Therefore, we are clear in our discussion that the survey was not designed to be representative of either the ECHO consortium or the U.S. population. See related question #2. 

29. Line 367: I wouldn’t refer to the fact that you are “speculating” about changes in environmental exposure as a strength of the study.

 Response: We agree. Wording was changed; see related question #19

30. Line 389: I don’t understand why a survey response rate couldn’t be calculated based on the number of people offered the survey and the number who completed it.

 Response: Because each cohort administered this survey using a different study protocol, it is difficult to know how many people were offered the survey, which would provide the denominator for a response rate. For example, some cohorts call participants 1-by-1, some “blast” the surveys to all participants, some mail surveys. This variation in distribution methods makes the calculation of a formal response rate very complicated/impossible. 

31. Line 392: what is meant by a “virtual” cohort study? Also, can this really be called national when the responses are dominated by rural women from NH?

 Response: See question #5; text was added (lines 21-23 and 94-100) to provide more context about the consortium. We agree about characterizing the response to this survey as ‘virtual’ or ‘national’ and have tempered this language on line 431. 

Reviewer #3 (R1) [did not provide overview comments]

32. Line 39-42: Recent research indicates that air pollution has returned to pre-pandemic levels as recently as February 2021 – please adjust to reflect these data or change the wording to reflect that they have rebounded.

 Response: wording has been updated accordingly

33. Line 121-125: The study is primarily coming from New Hampshire, which is a very racially homogenous (93% are white alone) state and low in overall population, comparatively speaking. This is an impressive study – as such, what is your rationale for reporting premature results from mostly NH alone, given the substantial generalizability issues and potential for bias? I worry that in trying to tease out racial effects, you’re instead looking at a very small group that could be experiencing heightened stress already because of being an extreme minority, as one example.

 Response: In all of our multivariable analyses, we included an indicator term for whether a respondent was part of the NH cohort or not. This term was rarely significant in analyses, leading us to believe that the influence of being from the NH cohort was not a significant driver of reported behavior change in most cases. Any differences in race/ethnicity were detected after adjusting for whether or not participants were from the NH cohort. 

34. Line 147-155: Are you saying here that you are comparing impacts from early COVID outbreaks to later stage COVID (e.g., post vaccine)? If so, it may benefit you to explicitly mention this, as there are many disparate factors that can impact these results.

 Response: we add more detail about why we chose to look at this cut-point (lines 176-177) (see response to question #25)

35. Line 181-185: It seems like stress and eating behaviors are a bidirectional relationship. Do you have some method to help control for this?

 Response: We agree. We point this out in our discussion (lines 383-385) that because both responses were recorded at the same time, it is not possible for us to infer directionality. 

36. Line 211-214: This is a small percentage of participants who are pregnant. My impression was that this was an area of interest, as pre and perinatal outcomes are highlighted as very important in the introduction.

 Response: We have removed some of this text in our introduction. While exposures during pregnancy as well as in early life are of interest, the focus (and the target) of this paper is exposures occurring among women of childbearing age, which is the majority (92%) of our respondents. 

37. Line 209-223: I am reminded of the many other stressors that were extremely likely co-occur during this same time in these states; namely, there has been a good deal of research identifying that political stress is matched alongside pandemic stress, with pandemic stress decreasing slightly over time while political stress remained static. There are major differences here by race, ethnicity, and occupation. Do you have any way of controlling for other stressors? Do the COVID questions specifically inquire about stress related to the outbreak of COVID?

 Response: The questions about stress specifically ask about COVID-related stress (additional text clarifying the question wording was added in lines 161-162; however, we cannot rule out that co-occurring stressors (e.g., political stress) are also contributing to reported symptoms. We have added a comment about this in the discussion (lines 390-393, see related question #47).

38. Line 255-265: Are these significant? If so, please specify.

 Response: P-values were added.

39. Line 267-278: Please specify which relationships are not significant.

 Response: P-values were added.

40. Line 276-278: Fascinating.

 Response: thank you

41. Line 283-287: These factors seem like they would be highly correlated. Is there a risk of collinearity here?

 Response: We agree that there is some collinearity, however not enough to prevent us from including these indicators in multivariable models so that we could consider independent effects. 

42. Line 280-293: I am confused by the decision to use percentages and p values separately and struggle to track what relationships are significant and to what magnitude, both in size of difference and significance. This undermines the strength of the paper.

 Response: We are not sure of a better way to report our findings beyond percent of participants who reported particular behaviors. The p-values associated with these percentages provide an indication of the likelihood of observing differentces as large as those found, assuming the null hypothesis of no difference. In Table 2, we present results as a function of odds ratios, which provides an additional way to consider our findings. 

43. Line 330-335: I am confused by this paragraph. Please try and clarify the message.

 Response: We added a clause at the start of the paragraph (line 360-361) to indicate that based on previous literature, we might draw inferences about the impact of the behavior changes in terms of environmental exposures. 

44. Line 337-344: Could this not be because of the sample sizes? I don’t think it’s fair to assume that these results can be used to anticipate subpopulations who are more likely to be exposed to a variety of chemicals, especially in the context of the complaints I have raised above.

 Response: We have tempered this response in light of your comments (see Conclusion, beginning line 431). 

45. Line 346-354: This is a great paragraph.

 Response: thank you

46. Line 362-368: While I do agree with most of this, this is not a diverse population in terms of geography, race, or ethnicity. This statement should be removed.

 Response: We feel that this sentence, which reads a “somewhat diverse population in terms of geography, race, and ethnicity” [emphasis added here] appropriately qualifies the claim regarding sample diversity. 

47. Line 370-389: I appreciate the candor by which the authors reflect on the limitations of the study. However, I feel it is a major flaw that you cannot truly account for COVID-19 related stress alone. My understanding of the paper is that you are assessing for stress during COVID-19, but not directly measuring stress as a result of COVID-19 alone. Is this a fair assessment? If so, I think this should be discussed in this section.

 Response: We have added text about the wording of the questions (see section beginning line 135) and a comment in the discussion about this (lines 390-393). See also question #37.

48. Line 391-403: I think the authors next step should be to analyze this same relationship in a complete population, rather than the limited scope identified in this paper.

 Response: Unfortunately, due to the demands of the ECHO-wide data collection protocol, not all ECHO participants have or will be asked the questions on this survey. Therefore, it won’t be possible to do what you suggest. 

Editorial Comments upon resubmission

49. Please remove any funding-related text from the manuscript.

 Response: Funding information was removed from the manuscript text. 

50. We note that your author list was updated during the revision process; please submit a ‘change in authorship’ form.

 Response: Actually, we did not add any coauthors in response to the review. We did update the order that their names appear, to reflect additional work/effort that we did as part of the review process. We interpreted this to mean that we do not need to submit a change in authorship form, as no new authors were added. Apologies for the confusion.

---

## [Decision Letter · Decision Letter 1]

2 Nov 2022

Characterizing changes in behaviors associated with chemical exposures during the COVID-19 pandemic

PONE-D-22-05387R1

Dear Dr. Herbstman,

We’re pleased to inform you that your manuscript has been judged scientifically suitable for publication and will be formally accepted for publication once it meets all outstanding technical requirements.

Kind regards,

Aaron Specht

Academic Editor

PLOS ONE

Additional Editor Comments (optional):

Reviewers' comments:

Reviewer's Responses to Questions

**Comments to the Author**

1. If the authors have adequately addressed your comments raised in a previous round of review and you feel that this manuscript is now acceptable for publication, you may indicate that here to bypass the “Comments to the Author” section, enter your conflict of interest statement in the “Confidential to Editor” section, and submit your "Accept" recommendation.

Reviewer #1: All comments have been addressed

Reviewer #3: All comments have been addressed

2. Is the manuscript technically sound, and do the data support the conclusions?

Reviewer #1: Yes

Reviewer #3: Partly

3. Has the statistical analysis been performed appropriately and rigorously? 

Reviewer #1: Yes

Reviewer #3: Yes

4. Have the authors made all data underlying the findings in their manuscript fully available?

Reviewer #1: Yes

Reviewer #3: Yes

5. Is the manuscript presented in an intelligible fashion and written in standard English?

Reviewer #1: Yes

Reviewer #3: Yes

6. Review Comments to the Author

Reviewer #1: I am satisfied with the authors' responses and manuscript changes in response to my comments on the first round of peer review. I commend the authors on a thorough revision and a manuscript that will be of interest to many readers.

Reviewer #3: Thank you for addressing the comments! This is a great paper. I do disagree with some statements; notably, the somewhat misleading decision to describe the population as diverse given its clear homogeneity. I also feel that the authors should wait until they have completed data collection before publishing. However, in its current form this draft is suitable for publication and I support it moving forward. Excellent work!

7. PLOS authors have the option to publish the peer review history of their article (what does this mean?). If published, this will include your full peer review and any attached files.

Reviewer #1: **Yes: **Aaron Reuben

Reviewer #3: **Yes: **Christian Hoover

---

## [Editor Report · Acceptance letter]

4 Jan 2023

PONE-D-22-05387R1 

Characterizing changes in behaviors associated with chemical exposures during the COVID-19 pandemic 

Dear Dr. Herbstman:

I'm pleased to inform you that your manuscript has been deemed suitable for publication in PLOS ONE. Congratulations! Your manuscript is now with our production department. 

Kind regards, 

on behalf of

Dr. Aaron Specht 

Academic Editor

PLOS ONE